# Can Decentralized Algorithms Outperform Centralized Algorithms? A Case Study for Decentralized Parallel Stochastic Gradient Descent

**Xiangru Lian**[†]**, Ce Zhang**[∗]**, Huan Zhang**[+]**, Cho-Jui Hsieh**[+]**, Wei Zhang**[#]**, and Ji Liu**[†♮]

[†]University of Rochester, [∗]ETH Zurich

[+]University of California, Davis, [#]IBM T. J. Watson Research Center, [♮]Tencent AI lab

xiangru@yandex.com, ce.zhang@inf.ethz.ch, victzhang@gmail.com,

chohsieh@ucdavis.edu, weiz@us.ibm.com, ji.liu.uwisc@gmail.com

## Abstract

Most distributed machine learning systems nowadays, including TensorFlow and CNTK, are built in a centralized fashion. One bottleneck of centralized algorithms lies on high communication cost on the central node. Motivated by this, we ask, *can decentralized algorithms be faster than its centralized counterpart?*

Although decentralized PSGD (D-PSGD) algorithms have been studied by the control community, existing analysis and theory do not show any advantage over centralized PSGD (C-PSGD) algorithms, simply assuming the application scenario where only the decentralized network is available. In this paper, we study a D-PSGD algorithm and provide the first theoretical analysis that indicates a regime in which decentralized algorithms might outperform centralized algorithms for distributed stochastic gradient descent. This is because D-PSGD has comparable total computational complexities to C-PSGD but requires much less communication cost on the busiest node. We further conduct an empirical study to validate our theoretical analysis across multiple frameworks (CNTK and Torch), different network configurations, and computation platforms up to 112 GPUs. On network configurations with low bandwidth or high latency, D-PSGD can be up to one order of magnitude faster than its well-optimized centralized counterparts.

## 1 Introduction

In the context of distributed machine learning, decentralized algorithms have long been treated as a *compromise* — when the underlying network topology does not allow centralized communication, one *has to* resort to decentralized communication, while, understandably, paying for the "cost of being decentralized". In fact, most distributed machine learning systems nowadays, including TensorFlow and CNTK, are built in a centralized fashion. But *can decentralized algorithms be faster than their centralized counterparts?* In this paper, we provide the first theoretical analysis, verified by empirical experiments, for a positive answer to this question.

We consider solving the following stochastic optimization problem

$$\min_{x \in \mathbb{R}^N} f(x) := \mathbb{E}_{\xi \sim \mathcal{D}} F(x; \xi), \tag{1}$$

where $\mathcal{D}$ is a predefined distribution and $\xi$ is a random variable usually referring to a data sample in machine learning. This formulation summarizes many popular machine learning models including deep learning [LeCun et al., 2015], linear regression, and logistic regression.

Parallel stochastic gradient descent (PSGD) methods are leading algorithms in solving large-scale machine learning problems such as deep learning [Dean et al., 2012, Li et al., 2014], matrix completion

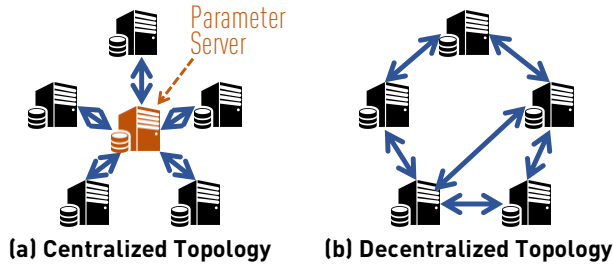

Figure 1: An illustration of different network topologies.

| Algorithm | communication complexity on the busiest node | computational complexity |
|---|---|---|
| C-PSGD (mini-batch SGD) | $O(n)$ | $O\left(\frac{n}{\epsilon} + \frac{1}{\epsilon^2}\right)$ |
| D-PSGD | $O\left(\text{Deg(network)}\right)$ | $O\left(\frac{n}{\epsilon} + \frac{1}{\epsilon^2}\right)$ |

Table 1: Comparison of C-PSGD and D-PSGD. The unit of the communication cost is the number of stochastic gradients or optimization variables. $n$ is the number of nodes. The computational complexity is the number of stochastic gradient evaluations we need to get a $\epsilon$-approximation solution, which is defined in (3).

[Recht et al., 2011, Zhuang et al., 2013] and SVM. Existing PSGD algorithms are mostly designed for centralized network topology, for example, the parameter server topology [Li et al., 2014], where there is a central node connected with multiple nodes as shown in Figure 1(a). The central node aggregates the stochastic gradients computed from all other nodes and updates the model parameter, for example, the weights of a neural network. The potential bottleneck of the centralized network topology lies on the communication traffic jam on the central node, because all nodes need to communicate with it concurrently iteratively. The performance will be significantly degraded when the network bandwidth is low.[1] These motivate us to study algorithms for *decentralized* topologies, where all nodes can only communicate with its neighbors and there is no such a central node, shown in Figure 1(b).

Although decentralized algorithms have been studied as consensus optimization in the control community and used for preserving data privacy [Ram et al., 2009a, Yan et al., 2013, Yuan et al., 2016], for the application scenario where only the decentralized network is available, it is still an open question **if decentralized methods could have advantages over centralized algorithms** in some scenarios in case both types of communication patterns are feasible — for example, on a supercomputer with thousands of nodes, *should we use decentralized or centralized communication?* Existing theory and analysis either do not make such comparison [Bianchi et al., 2013, Ram et al., 2009a, Srivastava and Nedic, 2011, Sundhar Ram et al., 2010] or implicitly indicate that decentralized algorithms were much worse than centralized algorithms in terms of computational complexity and total communication complexity [Aybat et al., 2015, Lan et al., 2017, Ram et al., 2010, Zhang and Kwok, 2014]. This paper gives a positive result for decentralized algorithms by studying a decentralized PSGD (D-PSGD) algorithm on the connected decentralized network. Our theory indicates that D-PSGD admits similar total computational complexity but requires much less communication for the busiest node. Table 1 shows a quick comparison between C-PSGD and D-PSGD with respect to the computation and communication complexity. Our contributions are:

- We theoretically justify the potential advantage of decentralizedalgorithms over centralized algorithms. Instead of treating decentralized algorithms as a compromise one has to make, we are the first to conduct a theoretical analysis that identifies cases in which decentralized algorithms can be faster than its centralized counterpart.
- We theoretically analyze the scalability behavior of decentralized SGD when more nodes are used. Surprisingly, we show that, when more nodes are available, decentralized algorithms can bring speedup, asymptotically linearly, with respect to computational complexity. To our best knowledge, this is the first speedup result related to decentralized algorithms.
- We conduct extensive empirical study to validate our theoretical analysis of D-PSGD and different C-PSGD variants (e.g., plain SGD, EASGD [Zhang et al., 2015]). We observe similar computational

complexity as our theory indicates; on networks with low bandwidth or high latency, D-PSGD can be up to $10\times$ faster than C-PSGD. Our result holds across multiple frameworks (CNTK and Torch), different network configurations, and computation platforms up to 112 GPUs. This indicates promising future direction in pushing the research horizon of machine learning systems from pure centralized topology to a more decentralized fashion.

**Definitions and notations**  Throughout this paper, we use following notation and definitions:

- $\|\cdot\|$ denotes the vector $\ell_2$ norm or the matrix spectral norm depending on the argument.
- $\|\cdot\|_F$ denotes the matrix Frobenius norm.
- $\nabla f(\cdot)$ denotes the gradient of a function $f$.
- $\mathbf{1}_n$ denotes the column vector in $\mathbb{R}^n$ with 1 for all elements.
- $f^*$ denotes the optimal solution of (1).
- $\lambda_i(\cdot)$ denotes the $i$-th largest eigenvalue of a matrix.

## 2   Related work

In the following, we use $K$ and $n$ to refer to the number of iterations and the number of nodes.

**Stochastic Gradient Descent (SGD)**  SGD is a powerful approach for solving large scale machine learning. The well known convergence rate of stochastic gradient is $O(1/\sqrt{K})$ for convex problems and $O(1/K)$ for strongly convex problems [Moulines and Bach, 2011, Nemirovski et al., 2009]. SGD is closely related to online learning algorithms, for example, Crammer et al. [2006], Shalev-Shwartz [2011], Yang et al. [2014]. For SGD on nonconvex optimization, an ergodic convergence rate of $O(1/\sqrt{K})$ is proved in Ghadimi and Lan [2013].

**Centralized parallel SGD**  For CENTRALIZED PARALLEL SGD (C-PSGD) algorithms, the most popular implementation is based on the parameter server, which is essentially the mini-batch SGD admitting a convergence rate of $O(1/\sqrt{Kn})$ [Agarwal and Duchi, 2011, Dekel et al., 2012, Lian et al., 2015], where in each iteration $n$ stochastic gradients are evaluated. In this implementation there is a parameter server communicating with all nodes. The linear speedup is implied by the convergence rate automatically. More implementation details for C-PSGD can be found in Chen et al. [2016], Dean et al. [2012], Li et al. [2014], Zinkevich et al. [2010]. The asynchronous version of centralized parallel SGD is proved to guarantee the linear speedup on all kinds of objectives (including convex, strongly convex, and nonconvex objectives) if the staleness of the stochastic gradient is bounded [Agarwal and Duchi, 2011, Feyzmahdavian et al., 2015, Lian et al., 2015, 2016, Recht et al., 2011, Zhang et al., 2016b,c].

**Decentralized parallel stochastic algorithms**  Decentralized algorithms do not specify any central node unlike centralized algorithms, and each node maintains its own local model but can only communicate with with its neighbors. Decentralized algorithms can usually be applied to any connected computational network. Lan et al. [2017] proposed a decentralized stochastic algorithm with computational complexities $O(n/\epsilon^2)$ for general convex objectives and $O(n/\epsilon)$ for strongly convex objectives. Sirb and Ye [2016] proposed an asynchronous decentralized stochastic algorithm ensuring complexity $O(n/\epsilon^2)$ for convex objectives. A similar algorithm to our D-PSGD in both synchronous and asynchronous fashion was studied in Ram et al. [2009a, 2010], Srivastava and Nedic [2011], Sundhar Ram et al. [2010]. The difference is that in their algorithm all node can only perform either communication or computation but not simultaneously. Sundhar Ram et al. [2010] proposed a stochastic decentralized optimization algorithm for constrained convex optimization and the algorithm can be used for non-differentiable objectives by using subgradients. Please also refer to Srivastava and Nedic [2011] for the subgradient variant. The analysis in Ram et al. [2009a, 2010], Srivastava and Nedic [2011], Sundhar Ram et al. [2010] requires the gradients of each term of the objective to be bounded by a constant. Bianchi et al. [2013] proposed a similar decentralized stochastic algorithm and provided a convergence rate for the consensus of the local models when the local models are bounded. The convergence to a solution was also provided by using central limit theorem, but the rate is unclear. HogWild++ [Zhang et al., 2016a] uses decentralized model parameters for parallel asynchronous SGD on multi-socket systems and shows that this algorithm empirically outperforms some centralized algorithms. Yet the convergence or the convergence rate is unclear. The common issue for these work above lies on that the speedup is unclear, that is, we do not know if decentralized algorithms (involving multiple nodes) can improve the efficiency of only using a single node.

**Other decentralized algorithms** In other areas including control, privacy and wireless sensing network, decentralized algorithms are usually studied for solving the consensus problem [Aysal et al., 2009, Boyd et al., 2005, Carli et al., 2010, Fagnani and Zampieri, 2008, Olfati-Saber et al., 2007, Schenato and Gamba, 2007]. Lu et al. [2010] proves a gossip algorithm to converge to the optimal solution for convex optimization. Mokhtari and Ribeiro [2016] analyzed decentralized SAG and SAGA algorithms for minimizing finite sum strongly convex objectives, but they are not shown to admit any speedup. The decentralized gradient descent method for convex and strongly convex problems was analyzed in Yuan et al. [2016]. Nedic and Ozdaglar [2009], Ram et al. [2009b] studied its subgradient variants. However, this type of algorithms can only converge to a ball of the optimal solution, whose diameter depends on the steplength. This issue was fixed by Shi et al. [2015] using a modified algorithm, namely EXTRA, that can guarantee to converge to the optimal solution. Wu et al. [2016] analyzed an asynchronous version of decentralized gradient descent with some modification like in Shi et al. [2015] and showed that the algorithm converges to a solution when $K \to \infty$. Aybat et al. [2015], Shi et al., Zhang and Kwok [2014] analyzed decentralized ADMM algorithms and they are not shown to have speedup. From all of these reviewed papers, it is still unclear if decentralized algorithms can have any advantage over their centralized counterparts.

## 3   Decentralized parallel stochastic gradient descent (D-PSGD)

---

**Algorithm 1** Decentralized Parallel Stochastic Gradient Descent (D-PSGD) on the $i$th node

---

**Require:** initial point $x_{0,i} = x_0$, step length $\gamma$, weight matrix $W$, and number of iterations $K$
1: **for** $k = 0, 1, 2, \ldots, K-1$ **do**
2:    Randomly sample $\xi_{k,i}$ from local data of the $i$-th node
3:    Compute the local stochastic gradient $\nabla F_i(x_{k,i}; \xi_{k,i})$ $\forall i$ on all nodes [a]
4:    Compute the neighborhood weighted average by fetching optimization variables from neighbors: $x_{k+\frac{1}{2},i} = \sum_{j=1}^{n} W_{ij} x_{k,j}$ [b]
5:    Update the local optimization variable $x_{k+1,i} \leftarrow x_{k+\frac{1}{2},i} - \gamma \nabla F_i(x_{k,i}; \xi_{k,i})$ [c]
6: **end for**
7: **Output:** $\frac{1}{n} \sum_{i=1}^{n} x_{K,i}$

---

[a]Note that the stochastic gradient computed in can be replaced with a mini-batch of stochastic gradients, which will not hurt our theoretical results.

[b]Note that the Line 3 and Line 4 can be run in parallel.

[c]Note that the Line 4 and step Line 5 can be exchanged. That is, we first update the local stochastic gradient into the local optimization variable, and then average the local optimization variable with neighbors. This does not hurt our theoretical analysis. When Line 4 is logically before Line 5, then Line 3 and Line 4 can be run in parallel. That is to say, if the communication time used by Line 4 is smaller than the computation time used by Line 3, the communication time can be completely hidden (it is overlapped by the computation time).

---

This section introduces the D-PSGD algorithm. We represent the decentralized communication topology with an undirected graph with weights: $(V, W)$. $V$ denotes the set of $n$ computational nodes: $V := \{1, 2, \cdots, n\}$. $W \in \mathbb{R}^{n \times n}$ is a symmetric doubly stochastic matrix, which means (i) $W_{ij} \in [0, 1], \forall i, j$, (ii) $W_{ij} = W_{ji}$ for all $i, j$, and (ii) $\sum_j W_{ij} = 1$ for all $i$. We use $W_{ij}$ to encode how much node $j$ can affect node $i$, while $W_{ij} = 0$ means node $i$ and $j$ are disconnected.

To design distributed algorithms on a decentralized network, we first distribute the data onto all nodes such that the original objective defined in (1) can be rewritten into

$$\min_{x \in \mathbb{R}^N} \quad f(x) = \frac{1}{n} \sum_{i=1}^{n} \underbrace{\mathbb{E}_{\xi \sim \mathcal{D}_i} F_i(x; \xi)}_{=:f_i(x)}. \tag{2}$$

There are two simple ways to achieve (2), both of which can be captured by our theoretical analysis and they both imply $F_i(\cdot; \cdot) = F(\cdot; \cdot), \forall i$.

**Strategy-1** All distributions $\mathcal{D}_i$'s are the same as $\mathcal{D}$, that is, all nodes can access a shared database;
**Strategy-2** $n$ nodes partition all data in the database and appropriately define a distribution for sampling local data, for example, if $\mathcal{D}$ is the uniform distribution over all data, $\mathcal{D}_i$ can be defined to be the uniform distribution over local data.

The D-PSGD algorithm is a synchronous parallel algorithm. All nodes are usually synchronized by a clock. Each node maintains its own local variable and runs the protocol in Algorithm 1 concurrently, which includes three key steps at iterate $k$:

- Each node computes the stochastic gradient $\nabla F_i(x_{k,i}; \xi_{k,i})^2$ using the current local variable $x_{k,i}$, where $k$ is the iterate number and $i$ is the node index;
- When the synchronization barrier is met, each node exchanges local variables with its neighbors and average the local variables it receives with its own local variable;
- Each node update its local variable using the average and the local stochastic gradient.

To view the D-PSGD algorithm from a global view, at iterate $k$, we define the concatenation of all local variables, random samples, stochastic gradients by matrix $X_k \in \mathbb{R}^{N \times n}$, vector $\xi_k \in \mathbb{R}^n$, and $\partial F(X_k, \xi_k)$, respectively:

$$X_k := [\ x_{k,1} \quad \cdots \quad x_{k,n}\ ] \in \mathbb{R}^{N \times n}, \quad \xi_k := [\ \xi_{k,1} \quad \cdots \quad \xi_{k,n}\ ]^\top \in \mathbb{R}^n,$$

$$\partial F(X_k, \xi_k) := [\ \nabla F_1(x_{k,1}; \xi_{k,1}) \quad \nabla F_2(x_{k,2}; \xi_{k,2}) \quad \cdots \quad \nabla F_n(x_{k,n}; \xi_{k,n})\ ] \in \mathbb{R}^{N \times n}.$$

Then the $k$-th iterate of Algorithm 1 can be viewed as the following update

$$X_{k+1} \leftarrow X_k W - \gamma \partial F(X_k; \xi_k).$$

We say the algorithm gives an $\epsilon$-approximation solution if

$$K^{-1} \left( \sum_{k=0}^{K-1} \mathbb{E} \left\| \nabla f \left( \tfrac{X_k \mathbf{1}_n}{n} \right) \right\|^2 \right) \leqslant \epsilon. \tag{3}$$

# 4 Convergence rate analysis

This section provides the analysis for the convergence rate of the D-PSGD algorithm. Our analysis will show that the convergence rate of D-PSGD w.r.t. iterations is similar to the C-PSGD (or mini-batch SGD) [Agarwal and Duchi, 2011, Dekel et al., 2012, Lian et al., 2015], but D-PSGD avoids the communication traffic jam on the parameter server.

To show the convergence results, we first define

$$\partial f(X_k) := [\ \nabla f_1(x_{k,1}) \quad \nabla f_2(x_{k,2}) \quad \cdots \quad \nabla f_n(x_{k,n})\ ] \in \mathbb{R}^{N \times n},$$

where functions $f_i(\cdot)$'s are defined in (2).

**Assumption 1.** *Throughout this paper, we make the following commonly used assumptions:*

*1. **Lipschitzian gradient:** All function $f_i(\cdot)$'s are with L-Lipschitzian gradients.*
*2. **Spectral gap:** Given the symmetric doubly stochastic matrix $W$, we define $\rho := (\max\{|\lambda_2(W)|, |\lambda_n(W)|\})^2$. We assume $\rho < 1$.*
*3. **Bounded variance:** Assume the variance of stochastic gradient $\mathbb{E}_{i \sim \mathcal{U}([n])} \mathbb{E}_{\xi \sim \mathcal{D}_i} \| \nabla F_i(x; \xi) - \nabla f(x) \|^2$ is bounded for any $x$ with $i$ uniformly sampled from $\{1, \ldots, n\}$ and $\xi$ from the distribution $\mathcal{D}_i$. This implies there exist constants $\sigma, \varsigma$ such that*

$$\mathbb{E}_{\xi \sim \mathcal{D}_i} \| \nabla F_i(x; \xi) - \nabla f_i(x) \|^2 \leqslant \sigma^2, \forall i, \forall x, \quad \mathbb{E}_{i \sim \mathcal{U}([n])} \| \nabla f_i(x) - \nabla f(x) \|^2 \leqslant \varsigma^2, \forall x.$$

*Note that if all nodes can access the shared database, then $\varsigma = 0$.*
*4. **Start from 0:** We assume $X_0 = 0$. This assumption simplifies the proof w.l.o.g.*

Let

$$D_1 := \left( \frac{1}{2} - \frac{9\gamma^2 L^2 n}{(1 - \sqrt{\rho})^2 D_2} \right), \quad D_2 := \left( 1 - \frac{18\gamma^2}{(1 - \sqrt{\rho})^2} n L^2 \right).$$

Under Assumption 1, we have the following convergence result for Algorithm 1.

**Theorem 1** (Convergence of Algorithm 1). *Under Assumption 1, we have the following convergence rate for Algorithm 1:*

$$\frac{1}{K} \left( \frac{1 - \gamma L}{2} \sum_{k=0}^{K-1} \mathbb{E} \left\| \frac{\partial f(X_k) \mathbf{1}_n}{n} \right\|^2 + D_1 \sum_{k=0}^{K-1} \mathbb{E} \left\| \nabla f \left( \frac{X_k \mathbf{1}_n}{n} \right) \right\|^2 \right)$$

$$\leqslant \frac{f(0) - f^*}{\gamma K} + \frac{\gamma L}{2n} \sigma^2 + \frac{\gamma^2 L^2 n \sigma^2}{(1 - \rho) D_2} + \frac{9\gamma^2 L^2 n \varsigma^2}{(1 - \sqrt{\rho})^2 D_2}.$$

Noting that $\frac{X_k \mathbf{1}_n}{n} = \frac{1}{n} \sum_{i=1}^{n} x_{k,i}$, this theorem characterizes the convergence of the average of all local optimization variables $x_{k,i}$. To take a closer look at this result, we appropriately choose the step length in Theorem 1 to obtain the following result:

**Corollary 2.** *Under the same assumptions as in Theorem 1, if we set* $\gamma = \frac{1}{2L + \sigma\sqrt{K/n}}$ [3]*, for Algorithm 1 we have the following convergence rate:*

$$\frac{\sum_{k=0}^{K-1} \mathbb{E} \left\| \nabla f \left( \frac{X_k \mathbf{1}_n}{n} \right) \right\|^2}{K} \leqslant \frac{8(f(0) - f^*)L}{K} + \frac{(8f(0) - 8f^* + 4L)\sigma}{\sqrt{Kn}}. \tag{4}$$

*if the total number of iterate $K$ is sufficiently large, in particular,*

$$K \geqslant \frac{4L^4 n^5}{\sigma^6 (f(0) - f^* + L)^2} \left( \frac{\sigma^2}{1 - \rho} + \frac{9\varsigma^2}{(1 - \sqrt{\rho})^2} \right)^2, \text{ and} \tag{5}$$

$$K \geqslant \frac{72 L^2 n^2}{\sigma^2 \left( 1 - \sqrt{\rho} \right)^2}. \tag{6}$$

This result basically suggests that the convergence rate for D-PSGD is $O\left( \frac{1}{K} + \frac{1}{\sqrt{nK}} \right)$, if $K$ is large enough. We highlight two key observations from this result:

**Linear speedup** When $K$ is large enough, the $\frac{1}{K}$ term will be dominated by the $\frac{1}{\sqrt{Kn}}$ term which leads to a $\frac{1}{\sqrt{nK}}$ convergence rate. It indicates that the total computational complexity[4] to achieve an $\epsilon$-approximation solution (3) is bounded by $O\left( \frac{1}{\epsilon^2} \right)$. Since the total number of nodes does not affect the total complexity, a single node only shares a computational complexity of $O\left( \frac{1}{n\epsilon^2} \right)$. Thus linear speedup can be achieved by D-PSGD asymptotically w.r.t. computational complexity.

**D-PSGD can be better than C-PSGD** Note that this rate is the same as C-PSGD (or mini-batch SGD with mini-batch size $n$) [Agarwal and Duchi, 2011, Dekel et al., 2012, Lian et al., 2015]. The advantage of D-PSGD over C-PSGD is to avoid the communication traffic jam. At each iteration, the *maximal* communication cost for every single node is $O$(the degree of the network) for D-PSGD, in contrast with $O(n)$ for C-PSGD. The degree of the network could be much smaller than $O(n)$, e.g., it could be $O(1)$ in the special case of a ring.

The key difference from most existing analysis for decentralized algorithms lies on that we do not use the boundedness assumption for domain or gradient or stochastic gradient. Those boundedness assumptions can significantly simplify the proof but lose some subtle structures in the problem.

The linear speedup indicated by Corollary 2 requires the total number of iteration $K$ is sufficiently large. The following special example gives a concrete bound of $K$ for the ring network topology.

**Theorem 3.** *(**Ring network**) Choose the steplength $\gamma$ in the same as Corollary 2 and consider the ring network topology with corresponding $W$ in the form of*

$$W = \begin{pmatrix} 1/3 & 1/3 & & & & & 1/3 \\ 1/3 & 1/3 & 1/3 & & & & \\ & 1/3 & 1/3 & \ddots & & & \\ & & & \ddots & & 1/3 & \\ & & & \ddots & \ddots & 1/3 & \\ & & & & 1/3 & 1/3 & 1/3 \\ 1/3 & & & & & 1/3 & 1/3 \end{pmatrix} \in \mathbb{R}^{n \times n}.$$

*Under Assumption 1, Algorithm 1 achieves the same convergence rate in* (4)*, which indicates a linear speedup can be achieved, if the number of involved nodes is bounded by*

- $n = O(K^{1/9})$*, if apply **strategy-1** distributing data ($\varsigma = 0$);*
- $n = O(K^{1/13})$*, if apply **strategy-2** distributing data ($\varsigma > 0$),*

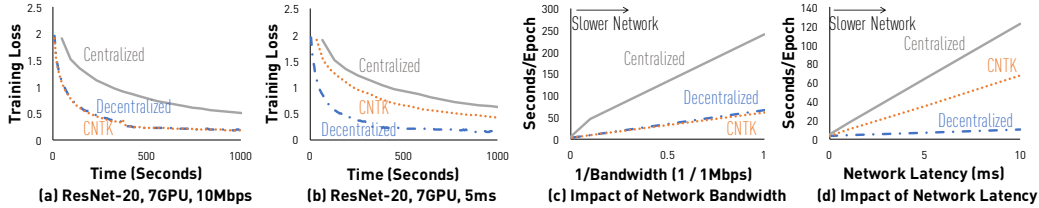

Figure 2: Comparison between D-PSGD and two centralized implementations (7 and 10 GPUs).

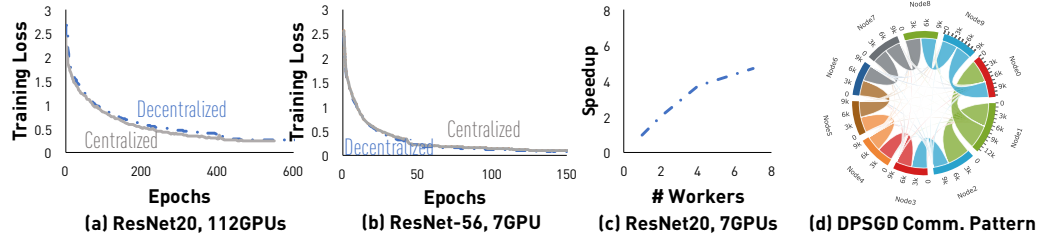

Figure 3: (a) Convergence Rate; (b) D-PSGD Speedup; (c) D-PSGD Communication Patterns.

*where the capital "O" swallows $\sigma, \varsigma, L$, and $f(0) - f^*$.*

This result considers a special decentralized network topology: ring network, where each node can only exchange information with its two neighbors. The linear speedup can be achieved up to $K^{1/9}$ and $K^{1/13}$ for different scenarios. These two upper bound can be improved potentially. This is the first work to show the speedup for decentralized algorithms, to the best of our knowledge.

In this section, we mainly investigate the convergence rate for the average of all local variables $\{x_{k,i}\}_{i=1}^n$. Actually one can also obtain a similar rate for each individual $x_{k,i}$, since all nodes achieve the consensus quickly, in particular, the running average of $\mathbb{E} \left\| \frac{1}{n} \sum_{i'=1}^n x_{k,i'} - x_{k,i} \right\|^2$ converges to 0 with a $O(1/K)$ rate, where the "$O$" swallows $n, \rho, \sigma, \varsigma, L$ and $f(0) - f^*$. See Theorem 6 for more details in Supplemental Material.

# 5 Experiments

We validate our theory with experiments that compare D-PSGD with other centralized implementations. We run experiments on clusters up to 112 GPUs and show that, on some network configurations, D-PSGD can outperform well-optimized centralized implementations by an order of magnitude.

## 5.1 Experiment setting

**Datasets and models** We evaluate D-PSGD on two machine learning tasks, namely (1) image classification, and (2) Natural Language Processing (NLP). For image classification we train ResNet [He et al., 2015] with different number of layers on CIFAR-10 [Krizhevsky, 2009]; for natural language processing, we train both proprietary and public dataset on a proprietary CNN model that we get from our industry partner [Feng et al., 2016, Lin et al., 2017, Zhang et al., 2017].

**Implementations and setups** We implement D-PSGD on two different frameworks, namely Microsoft CNTK and Torch. We evaluate four SGD implementations:

1. **CNTK.** We compare with the standard CNTK implementation of synchronous SGD. The implementation is based on MPI's AllReduce primitive.
2. **Centralized.** We implemented the standard parameter server-based synchronous SGD using MPI. One node will serve as the parameter server in our implementation.
3. **Decentralized.** We implemented our D-PSGD algorithm using MPI within CNTK.
4. **EASGD.** We compare with the standard EASGD implementation of Torch.

All three implementations are compiled with gcc 7.1, cuDNN 5.0, OpenMPI 2.1.1. We fork from CNTK after commit `57d7b9d` and enable distributed minibatch reading for all of our experiments.

During training, we keep the local batch size of each node the same as the reference configurations provided by CNTK. We tune learning rate for each SGD variant and report the best configuration.

**Machines/Clusters** We conduct experiments on three different machines/clusters:

1. **7GPUs.** A single local machine with 8 GPUs, each of which is a Nvidia TITAN Xp.
2. **10GPUs.** 10 `p2.xlarge` EC2 instances, each of which has one Nvidia K80 GPU.
3. **16GPUs.** 16 local machines, each of which has two Xeon E5-2680 8-core processors and a NVIDIA K20 GPU. Machines are connected by Gigabit Ethernet in this case.
4. **112GPUs.** 4 `p2.16xlarge` and 6 `p2.8xlarge` EC2 instances. Each `p2.16xlarge` (resp. `p2.8xlarge`) instance has 16 (resp. 8) Nvidia K80 GPUs.

In all of our experiments, we use each GPU as a node.

## 5.2 Results on CNTK

**End-to-end performance** We first validate that, under certain network configurations, D-PSGD converges faster, in wall-clock time, to a solution that has the same quality of centralized SGD. Figure 2(a, b) and Figure 3(a) shows the result of training ResNet20 on 7GPUs. We see that D-PSGD converges faster than both centralized SGD competitors. This is because when the network is slow, both centralized SGD competitors take more time per epoch due to communication overheads. Figure 3(a, b) illustrates the convergence with respect to the number of epochs, and D-PSGD shows similar convergence rate as centralized SGD even with 112 nodes.

**Speedup** The end-to-end speedup of D-PSGD over centralized SGD highly depends on the underlying network. We use the `tc` command to manually vary the network bandwidth and latency and compare the wall-clock time that all three SGD implementations need to finish one epoch.

Figure 2(c, d) shows the result. We see that, when the network has high bandwidth and low latency, not surprisingly, all three SGD implementations have similar speed. This is because in this case, the communication is never the system bottleneck. However, when the bandwidth becomes smaller (Figure 2(c)) or the latency becomes higher (Figure 2(d)), both centralized SGD implementations slow down significantly. In some cases, D-PSGD can be even one order of magnitude faster than its centralized competitors. Compared with **Centralized** (implemented with a parameter server), D-PSGD has more balanced communication patterns between nodes and thus outperforms **Centralized** in low-bandwidth networks; compared with **CNTK** (implemented with AllReduce), D-PSGD needs fewer number of communications between nodes and thus outperforms **CNTK** in high-latency networks. Figure 3(c) illustrates the communication between nodes for one run of D-PSGD.

We also vary the number of GPUs that D-PSGD uses and report the speed up over a single GPU to reach the same loss. Figure 3(b) shows the result on a machine with 7GPUs. We see that, up to 4 GPUs, D-PSGD shows near linear speed up. When all seven GPUs are used, D-PSGD achieves up to $5\times$ speed up. This subliner speed up for 7 GPUs is due to the synchronization cost but also that our machine only has 4 PCIe channels and thus more than two GPUs will share PCIe bandwidths.

## 5.3 Results on Torch

Due to the space limitation, the results on Torch can be found in Supplement Material.

## 6 Conclusion

This paper studies the D-PSGD algorithm on the decentralized computational network. We prove that D-PSGD achieves the same convergence rate (or equivalently computational complexity) as the C-PSGD algorithm, but outperforms C-PSGD by avoiding the communication traffic jam. To the best of our knowledge, this is the first work to show that decentralized algorithms admit the linear speedup and can outperform centralized algorithms.

**Limitation and Future Work** The potential limitation of D-PSGD lies on the cost of synchronization. Breaking the synchronization barrier could make the decentralize algorithms even more efficient, but requires more complicated analysis. We will leave this direction for the future work.

On the system side, one future direction is to deploy D-PSGD to larger clusters beyond 112 GPUs and one such environment is state-of-the-art supercomputers. In such environment, we envision D-PSGD to be one necessary building blocks for multiple "centralized groups" to communicate. It is also interesting to deploy D-PSGD to mobile environments.

**Acknowledgements** Xiangru Lian and Ji Liu are supported in part by NSF CCF1718513. Ce Zhang gratefully acknowledge the support from the Swiss National Science Foundation NRP 75 407540_167266, IBM Zurich, Mercedes-Benz Research & Development North America, Oracle Labs, Swisscom, Chinese Scholarship Council,

the Department of Computer Science at ETH Zurich, the GPU donation from NVIDIA Corporation, and the cloud computation resources from Microsoft Azure for Research award program. Huan Zhang and Cho-Jui Hsieh acknowledge the support of NSF IIS-1719097 and the TACC computation resources.

## Footnotes

[1]There has been research in how to accommodate this problem by having multiple parameter servers communicating with efficient MPI ALLREDUCE primitives. As we will see in the experiments, these methods, on the other hand, might suffer when the network latency is high.

[2]It can be easily extended to mini-batch stochastic gradient descent.

[3] In Theorem 1 and Corollary 2, we choose the constant steplength for simplicity. Using the diminishing steplength $O(\sqrt{n/k})$ can achieve a similar convergence rate by following the proof procedure in this paper. For convex objectives, D-PSGD could be proven to admit the convergence rate $O(1/\sqrt{nK})$ which is consistent with the non-convex case. For strongly convex objectives, the convergence rate for D-PSGD could be improved to $O(1/nK)$ which is consistent with the rate for C-PSGD.

[4] The complexity to compute a single stochastic gradient counts 1.

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
