[Supplementary Material]

# Supplemental Materials: Results on Torch

The following comparison is based on the implementation using Torch.

We provide results for the experiment of D-PSGD and EASGD. For this set of experiments we use a 32-layer residual network and CIFAR-10 dataset. We use up to 16 machines, and each machine includes two Xeon E5-2680 8-core processors and a NVIDIA K20 GPU. Worker machines are connected in a logical ring as described in Theorem 3. Connections between D-PSGD nodes are made via TCP socks, and EASGD uses MPI for communication. Because D-PSGD do not have a centralized model, we average all models from different machines as our final model to evaluate. In practical training, this only needs to be done after the last epoch with an all-reduce operation. For EASGD, we evaluate the central model on the parameter server.

One remarkable feature of this experiment is that we use inexpensive Gigabit Ethernet to connect all machines, and we are able to practically observe network congestion with centralized parameter server approach, even with a relatively small (ResNet-32) model. Although in practice, network with much higher bandwidth are available (e.g., InfiniBand), we also want to use larger model or more machines, so that network bandwidth can always become a bottleneck. We practically show that D-PSGD has better scalability than centralized approaches when network bandwidth becomes a constraint.

**Comparison to EASGD** Elastic Averaging SGD (EASGD) [Zhang et al., 2015] is an improved parameter server approach that outperforms traditional parameter server [Dean et al., 2012]. It makes each node perform more exploration by allowing local parameters to fluctuate around the central variable. We add ResNet-32 [He et al., 2016] with CIFAR-10 into the EASGD's Torch experiment code[5] and also implement our algorithm in Torch. Both algorithms run at the same speed on a single GPU so there is no implementation bias. Unlike the previous experiment which uses high bandwidth PCI-e or 10Gbits network for inter-GPU communication, we use 9 physical machines (1 as parameter server) with a single K20 GPU each, connected by inexpensive Gigabit Ethernet. For D-PSGD we use a logical ring connection between nodes as in Theorem 3. For EASGD we set moving rate $\beta = 0.9$ and use its momentum variant (EAMSGD). For both algorithms we set learning rate to 0.1, momentum to 0.9. $\tau = \{1, 4, 16\}$ is a hyper-parameter in EASGD controlling the number of mini-batches before communicating with the server.

(a) Iteration vs Training Loss      (b) Time vs Training Loss

Figure 4: Convergence comparison between D-PSGD and EAMSGD (EASGD's momentum variant).

Figure 4 shows that D-PSGD outperforms EASGD with a large margin in this setting. EASGD with $\tau = 1$ has good convergence, but its large bandwidth requirement saturates the network and slows down nodes. When $\tau = 4, 16$ EASGD converges slower than D-PSGD as there is less communication. D-PSGD allows more communication in an efficient way without reaching the network bottleneck. Moreover, D-PSGD is synchronous and shows less convergence fluctuation comparing with EASGD.

**Accuracy comparison with EASGD** We have shown the training loss comparison between D-PSGD and EASGD, and we now show additional figures comparing training error and test error in our experiment, as in Figure 5 and 6. We observe similar results as we have seen in section 6; D-PSGD can achieve good accuracy noticeably faster than EASGD.

(a) Iteration vs Training Error                (b) Time vs Training Error

Figure 5: Training Error comparison between D-PSGD and EAMSGD (EASGD's momentum variant)

(a) Iteration vs Test Error                (b) Time vs Test Error

Figure 6: Test Error comparison between D-PSGD and EAMSGD (EASGD's momentum variant)

**Scalability of D-PSGD**  In this experiment, we run D-PSGD on 1, 4, 8, 16 machines and compare convergence speed and error. For experiments involving 16 machines, each machine also connects to one additional machine which has the largest topological distance on the ring besides its two logical neighbours. We found that this can help information flow and get better convergence.

In Figure 10, 11 and 12 we can observe that D-PSGD scales very well when the number of machines is growing. Also, comparing with the single machine SGD, D-PSGD has minimum overhead; we measure the per-epoch training time only increases by 3% comparing to single machine SGD, but D-PSGD's convergence speed is much faster. To reach a training loss of 0.2, we need about 80 epochs with 1 machine, 20 epochs with 4 machines, 10 epochs with 8 machines and only 5 epochs with 16 machines. The observed linear speedup justifies the correctness of our theory.

(a) Iteration vs Training Loss                (b) Time vs Training Loss

Figure 7: Training Loss comparison between D-PSGD on 1, 4, 8 and 16 machines

**Generalization ability of D-PSGD**  In our previous experiments we set the learning rate to fixed 0.1. To complete Residual network training, we need to decrease the learning rate after some epochs. We follow the learning rate schedule in ResNet paper [He et al., 2016], and decrease the learning rate

(a) Iteration vs Training Error  (b) Time vs Training Error

Figure 8: Training Error comparison between D-PSGD on 1, 4, 8 and 16 machines

(a) Iteration vs Test Error  (b) Time vs Test Error

Figure 9: Test Error comparison between D-PSGD on 1, 4, 8 and 16 machines

to 0.01 at epoch 80. We observe training/test loss and error, as shown in figure 10, 11 and 12. For D-PSGD, we can tune a better learning rate schedule, but parameter tuning is not the focus of our experiments; rather, we would like to see if D-PSGD can achieve the same best ResNet accuracy as reported by the literature.

(a) Iteration vs Training Loss  (b) Time vs Training Loss

Figure 10: Training Loss comparison between D-PSGD on 1, 4, 8 and 16 machines

The test error of D-PSGD after 160 epoch is 0.0715, 0.0746 and 0.0735, for 4, 8 and 16 machines, respectively. He et al. [2016] reports 0.0751 error for the same 32-layer residual network, and we can reliably outperform the reported error level regardless of different numbers of machines used. Thus, D-PSGD does not negatively affect (or perhaps helps) generalization.

**Network utilization** During the experiment, we measure the network bandwidth on each machine. Because every machine is identical on the network, the measured bandwidth are the same on each machines. For experiment with 4 and 8 machines, the required bandwidth is about 22 MB/s. With 16 machines the required bandwidth is about 33 MB/s because we have an additional link. The required bandwidth is related to GPU performance; if GPU can compute each minibatch faster, the required

(a) Iteration vs Training Error　　　　　　　(b) Time vs Training Error

Figure 11: Training Error comparison between D-PSGD on 1, 4, 8 and 16 machines

(a) Iteration vs Test Error　　　　　　　(b) Time vs Test Error

Figure 12: Test Error comparison between D-PSGD on 1, 4, 8 and 16 machines

bandwidth also increases proportionally. Considering the practical bandwidth of Gigabit Ethernet is about 100 ~120 MB/s, Our algorithm can handle a 4 ~5 times faster GPU (or GPUs) easily, even with an inexpensive gigabit connection.

Because our algorithm is synchronous, we desire each node to compute each minibatch roughly within the same time. If each machine has different computation power, we can use different minibatch sizes to compensate the speed difference, or allow faster machines to make more than 1 minibatch before synchronization.

**Industrial benchmark**

In this section, we evaluate the effectiveness of our algorithm on IBM Watson Natural Language Classifier (NLC) workload. IBM Watson Natural Language Classifier (NLC) service, IBM's most popular cognitive service offering, is used by thousands of enterprise-level clients around the globe. The NLC task is to classify input sentences into a target category in a predefined label set. NLC has been extensively used in many practical applications, including sentiment analysis, topic classification, and question classification. At the core of NLC training is a CNN model that has a word-embedding lookup table layer, a convolutional layer and a fully connected layer with a softmax output layer. NLC is implemented using the Torch open-source deep learning framework.

**Methodology** We use two datasets in our evaluation. The first dataset Joule is an in-house customer dataset that has 2.5K training samples, 1K test samples, and 311 different classes. The second dataset Yelp, which is a public dataset, has 500K training samples, 2K test samples and 5 different classes. The experiments are conducted on an IBM Power server, which has 40 IBM P8 cores, each core is 4-way SMP with clock frequence of 2GHz. The server has 128GB memory and is equipped with 8 K80 GPUs. DataParallelTable (DPT) is a NCCL-basedNvidia module in Torch that can leverage multiple GPUs to carry out centralized parallel SGD algorithm. NCCL is an all-reduce based implementation. We implemented the decentralized SGD algorithm in the NLC product. We now compare the convergence rate of centralized SGD (i.e. DPT) and our decentralized SGD implementation.

**Convergence results and test accuracy** First, we examine the Joule dataset. We use 8 nodes and each node calculates with a mini-batch size of 2 and the entire run passes through 200 epochs. Figure 13 shows that centralized SGD algorithm and decentralized SGD algorithm achieve similar training loss (0.96) at roughly same convergence rate. Figure 14 shows that centralized SGD algorithm and decentralized SGD algorithm achieve similar testing error (43%). In the meantime, the communication cost is reduced by 3X in decentralized SGD case compared to the centralized SGD algorithm. Second, we examine the Yelp dataset. We use 8 nodes and each node calculates with a mini-batch size of 32 and the entire run passes through 20 epochs. Figure 13 shows that centralized SGD algorithm and decentralized SGD algorithm achieve similar training loss (0.86). Figure 14 shows that centralized SGD algorithm and decentralized SGD algorithm achieve similar testing error (39%).

Figure 13: Training loss on Joule dataset

Figure 14: Test error on Joule dataset

Figure 15: Training loss on Yelp dataset

Figure 16: Test error on Yelp dataset

# Supplemental Materials: Proofs

We provide the proof to all theoretical results in this paper in this section.

**Lemma 4.** *Under Assumption 1 we have*

$$\left\|\frac{\mathbf{1}_n}{n} - W^k e_i\right\|^2 \le \rho^k, \quad \forall i \in \{1, 2, \ldots, n\}, k \in \mathbb{N}.$$

*Proof.* Let $W^\infty := \lim_{k\to\infty} W^k$. Note that from Assumption 1-2 we have $\frac{\mathbf{1}_n}{n} = W^\infty e_i, \forall i$ since $W$ is doubly stochastic and $\rho < 1$. Thus

$$\begin{aligned}
\left\|\frac{\mathbf{1}_n}{n} - W^k e_i\right\|^2 &= \|(W^\infty - W^k)e_i\|^2 \\
&\le \|W^\infty - W^k\|^2 \|e_i\|^2 \\
&= \|W^\infty - W^k\|^2 \\
&\le \rho^k,
\end{aligned}$$

where the last step comes from the diagonalizability of $W$, completing the proof. $\square$

**Lemma 5.** *We have the following inequality under Assumption 1:*

$$\mathbb{E}\|\partial f(X_j)\|^2 \le \sum_{h=1}^n 3\mathbb{E}L^2 \left\|\frac{\sum_{i'=1}^n x_{j,i'}}{n} - x_{j,h}\right\|^2 + 3n\varsigma^2 + 3\mathbb{E}\left\|\nabla f\left(\frac{X_j \mathbf{1}_n}{n}\right)\mathbf{1}_n^\top\right\|^2, \forall j.$$

*Proof.* We consider the upper bound of $\mathbb{E}\|\partial f(X_j)\|^2$ in the following:

$$\mathbb{E}\|\partial f(X_j)\|^2$$

$$\begin{aligned}
\le &\, 3\mathbb{E}\left\|\partial f(X_j) - \partial f\left(\frac{X_j \mathbf{1}_n}{n}\mathbf{1}_n^\top\right)\right\|^2 \\
&+ 3\mathbb{E}\left\|\partial f\left(\frac{X_j \mathbf{1}_n}{n}\mathbf{1}_n^\top\right) - \nabla f\left(\frac{X_j \mathbf{1}_n}{n}\right)\mathbf{1}_n^\top\right\|^2 \\
&+ 3\mathbb{E}\left\|\nabla f\left(\frac{X_j \mathbf{1}_n}{n}\right)\mathbf{1}_n^\top\right\|^2 \\
\overset{\text{(Assumption 1-3)}}{\le} &\, 3\mathbb{E}\left\|\partial f(X_j) - \partial f\left(\frac{X_j \mathbf{1}_n}{n}\mathbf{1}_n^\top\right)\right\|_F^2 \\
&+ 3n\varsigma^2 \\
&+ 3\mathbb{E}\left\|\nabla f\left(\frac{X_j \mathbf{1}_n}{n}\right)\mathbf{1}_n^\top\right\|^2 \\
\overset{\text{(Assumption 1-1)}}{\le} &\, \sum_{h=1}^n 3\mathbb{E}L^2 \left\|\frac{\sum_{i'=1}^n x_{j,i'}}{n} - x_{j,h}\right\|^2 + 3n\varsigma^2 + 3\mathbb{E}\left\|\nabla f\left(\frac{X_j \mathbf{1}_n}{n}\right)\mathbf{1}_n^\top\right\|^2.
\end{aligned}$$

This completes the proof. $\square$

***Proof to Theorem 1.*** We start form $f\left(\frac{X_{k+1}\mathbf{1}_n}{n}\right)$:

$$\mathbb{E}f\left(\frac{X_{k+1}\mathbf{1}_n}{n}\right)$$

$$= \mathbb{E}f\left(\frac{X_k W \mathbf{1}_n}{n} - \gamma\frac{\partial F(X_k; \xi_k)\mathbf{1}_n}{n}\right)$$

$$\overset{\text{(Assumption 1-2)}}{=} \mathbb{E}f\left(\frac{X_k \mathbf{1}_n}{n} - \gamma\frac{\partial F(X_k; \xi_k)\mathbf{1}_n}{n}\right)$$

$$\leqslant \mathbb{E}f\left(\frac{X_k \mathbf{1}_n}{n}\right) - \gamma \mathbb{E}\left\langle \nabla f\left(\frac{X_k \mathbf{1}_n}{n}\right), \frac{\partial f(X_k)\mathbf{1}_n}{n}\right\rangle$$

$$+ \frac{\gamma^2 L}{2}\mathbb{E}\left\|\frac{\sum_{i=1}^{n} \nabla F_i(x_{k,i}; \xi_{k,i})}{n}\right\|^2. \tag{7}$$

Note that for the last term we can split it into two terms:

$$\mathbb{E}\left\|\frac{\sum_{i=1}^{n} \nabla F_i(x_{k,i}; \xi_{k,i})}{n}\right\|^2 = \mathbb{E}\left\|\frac{\sum_{i=1}^{n} \nabla F_i(x_{k,i}; \xi_{k,i}) - \sum_{i=1}^{n} \nabla f_i(x_{k,i})}{n} + \frac{\sum_{i=1}^{n} \nabla f_i(x_{k,i})}{n}\right\|^2$$

$$= \mathbb{E}\left\|\frac{\sum_{i=1}^{n} \nabla F_i(x_{k,i}; \xi_{k,i}) - \sum_{i=1}^{n} \nabla f_i(x_{k,i})}{n}\right\|^2$$

$$+ \mathbb{E}\left\|\frac{\sum_{i=1}^{n} \nabla f_i(x_{k,i})}{n}\right\|^2$$

$$+ \mathbb{E}\left\langle \frac{\sum_{i=1}^{n} \nabla F_i(x_{k,i}; \xi_{k,i}) - \sum_{i=1}^{n} \nabla f_i(x_{k,i})}{n}, \frac{\sum_{i=1}^{n} \nabla f_i(x_{k,i})}{n}^2\right\rangle$$

$$= \mathbb{E}\left\|\frac{\sum_{i=1}^{n} \nabla F_i(x_{k,i}; \xi_{k,i}) - \sum_{i=1}^{n} \nabla f_i(x_{k,i})}{n}\right\|^2$$

$$+ \mathbb{E}\left\|\frac{\sum_{i=1}^{n} \nabla f_i(x_{k,i})}{n}\right\|^2$$

$$+ \mathbb{E}\left\langle \frac{\sum_{i=1}^{n} \mathbb{E}_{\xi_{k,i}} \nabla F_i(x_{k,i}; \xi_{k,i}) - \sum_{i=1}^{n} \nabla f_i(x_{k,i})}{n}, \frac{\sum_{i=1}^{n} \nabla f_i(x_{k,i})}{n}^2\right\rangle$$

$$= \mathbb{E}\left\|\frac{\sum_{i=1}^{n} \nabla F_i(x_{k,i}; \xi_{k,i}) - \sum_{i=1}^{n} \nabla f_i(x_{k,i})}{n}\right\|^2$$

$$+ \mathbb{E}\left\|\frac{\sum_{i=1}^{n} \nabla f_i(x_{k,i})}{n}\right\|^2.$$

Then it follows from (7) that

$$\mathbb{E}f\left(\frac{X_{k+1}\mathbf{1}_n}{n}\right)$$

$$\leqslant \mathbb{E}f\left(\frac{X_k \mathbf{1}_n}{n}\right) - \gamma \mathbb{E}\left\langle \nabla f\left(\frac{X_k \mathbf{1}_n}{n}\right), \frac{\partial f(X_k)\mathbf{1}_n}{n}\right\rangle$$

$$+ \frac{\gamma^2 L}{2}\mathbb{E}\left\|\frac{\sum_{i=1}^{n} \nabla F_i(x_{k,i}; \xi_{k,i}) - \sum_{i=1}^{n} \nabla f_i(x_{k,i})}{n}\right\|^2$$

$$+ \frac{\gamma^2 L}{2}\mathbb{E}\left\|\frac{\sum_{i=1}^{n} \nabla f_i(x_{k,i})}{n}\right\|^2. \tag{8}$$

For the second last term we can bound it using $\sigma$:

$$\frac{\gamma^2 L}{2}\mathbb{E}\left\|\frac{\sum_{i=1}^{n} \nabla F_i(x_{k,i}; \xi_{k,i}) - \sum_{i=1}^{n} \nabla f_i(x_{k,i})}{n}\right\|^2$$

$$= \frac{\gamma^2 L}{2n^2}\sum_{i=1}^{n}\mathbb{E}\|\nabla F_i(x_{k,i}; \xi_{k,i}) - \nabla f_i(x_{k,i})\|^2$$

$$+ \frac{\gamma^2 L}{n^2}\sum_{i=1}^{n}\sum_{i'=i+1}^{n}\mathbb{E}\langle \nabla F_i(x_{k,i}; \xi_{k,i}) - \nabla f_i(x_{k,i}), \nabla F_{i'}(x_{k,i'}; \xi_{k,i'}) - \nabla f_{i'}(x_{k,i'})\rangle$$

$$= \frac{\gamma^2 L}{2n^2}\sum_{i=1}^{n}\mathbb{E}\|\nabla F_i(x_{k,i}; \xi_{k,i}) - \nabla f_i(x_{k,i})\|^2$$

$$+ \frac{\gamma^2 L}{n^2} \sum_{i=1}^{n} \sum_{i'=i+1}^{n} \mathbb{E}\langle \nabla F_i(x_{k,i}; \xi_{k,i}) - \nabla f_i(x_{k,i}), \mathbb{E}_{\xi_{k,i'}} \nabla F_{i'}(x_{k,i'}; \xi_{k,i'}) - \nabla f_{i'}(x_{k,i'})\rangle$$

$$= \frac{\gamma^2 L}{2n^2} \sum_{i=1}^{n} \mathbb{E}\|\nabla F_i(x_{k,i}; \xi_{k,i}) - \nabla f_i(x_{k,i})\|^2$$

$$\leqslant \frac{\gamma^2 L}{2n} \sigma^2,$$

where the last step comes from Assumption 1-3.

Thus it follows from (8):

$$\mathbb{E}f\left(\frac{X_{k+1}\mathbf{1}_n}{n}\right)$$

$$\leqslant \mathbb{E}f\left(\frac{X_k\mathbf{1}_n}{n}\right) - \gamma\mathbb{E}\left\langle \nabla f\left(\frac{X_k\mathbf{1}_n}{n}\right), \frac{\partial f(X_k)\mathbf{1}_n}{n}\right\rangle + \frac{\gamma^2 L}{2}\frac{\sigma^2}{n}$$

$$+ \frac{\gamma^2 L}{2}\mathbb{E}\left\|\frac{\sum_{i=1}^{n}\nabla f_i(x_{k,i})}{n}\right\|^2$$

$$= \mathbb{E}f\left(\frac{X_k\mathbf{1}_n}{n}\right) - \frac{\gamma - \gamma^2 L}{2}\mathbb{E}\left\|\frac{\partial f(X_k)\mathbf{1}_n}{n}\right\|^2 - \frac{\gamma}{2}\mathbb{E}\left\|\nabla f\left(\frac{X_k\mathbf{1}_n}{n}\right)\right\|^2 + \frac{\gamma^2 L}{2}\frac{\sigma^2}{n}$$

$$+ \underbrace{\frac{\gamma}{2}\mathbb{E}\left\|\nabla f\left(\frac{X_k\mathbf{1}_n}{n}\right) - \frac{\partial f(X_k)\mathbf{1}_n}{n}\right\|^2}_{=:T_1}, \tag{9}$$

where the last step comes from $2\langle a, b\rangle = \|a\|^2 + \|b\|^2 - \|a-b\|^2$.

We then bound $T_1$:

$$T_1 = \mathbb{E}\left\|\nabla f\left(\frac{X_k\mathbf{1}_n}{n}\right) - \frac{\partial f(X_k)\mathbf{1}_n}{n}\right\|^2$$

$$\leqslant \frac{1}{n}\sum_{i=1}^{n}\mathbb{E}\left\|\nabla f_i\left(\frac{\sum_{i'=1}^{n}x_{k,i'}}{n}\right) - \nabla f_i(x_{k,i})\right\|^2$$

$$\overset{\text{(Assumption 1-1)}}{\leqslant} \frac{L^2}{n}\sum_{i=1}^{n}\mathbb{E}\underbrace{\left\|\frac{\sum_{i'=1}^{n}x_{k,i'}}{n} - x_{k,i}\right\|^2}_{=:Q_{k,i}}, \tag{10}$$

where we define $Q_{k,i}$ as the squared distance of the local optimization variable on the $i$-th node from the averaged local optimization variables on all nodes.

In order to bound $T_1$ we bound $Q_{k,i}$'s as the following:

$$Q_{k,i} = \mathbb{E}\left\|\frac{\sum_{i'=1}^{n}x_{k,i'}}{n} - x_{k,i}\right\|^2$$

$$= \mathbb{E}\left\|\frac{X_k\mathbf{1}_n}{n} - X_ke_i\right\|^2$$

$$= \mathbb{E}\left\|\frac{X_{k-1}W\mathbf{1}_n - \gamma\partial F(X_{k-1}; \xi_{k-1})\mathbf{1}_n}{n} - (X_{k-1}We_i - \gamma\partial F(X_{k-1}; \xi_{k-1})e_i)\right\|^2$$

$$= \mathbb{E}\left\|\frac{X_{k-1}\mathbf{1}_n - \gamma\partial F(X_{k-1}; \xi_{k-1})\mathbf{1}_n}{n} - (X_{k-1}We_i - \gamma\partial F(X_{k-1}; \xi_{k-1})e_i)\right\|^2$$

$$= \mathbb{E}\left\|\frac{X_0\mathbf{1}_n - \sum_{i=0}^{k-1}\gamma\partial F(X_i; \xi_i)\mathbf{1}_n}{n} - \left(X_0W^ke_i - \sum_{j=0}^{k-1}\gamma\partial F(X_j; \xi_j)W^{k-j-1}e_i\right)\right\|^2$$

$$=\mathbb{E}\left\|X_0\left(\frac{\mathbf{1}_n}{n}-W^k e_i\right)-\sum_{j=0}^{k-1}\gamma\partial F(X_j;\xi_j)\left(\frac{\mathbf{1}_n}{n}-W^{k-j-1}e_i\right)\right\|^2$$

$$\overset{\text{(Assumption 1-4)}}{=}\mathbb{E}\left\|\sum_{j=0}^{k-1}\gamma\partial F(X_j;\xi_j)\left(\frac{\mathbf{1}_n}{n}-W^{k-j-1}e_i\right)\right\|^2$$

$$=\gamma^2\mathbb{E}\left\|\sum_{j=0}^{k-1}\partial F(X_j;\xi_j)\left(\frac{\mathbf{1}_n}{n}-W^{k-j-1}e_i\right)\right\|^2$$

$$\leqslant 2\gamma^2\underbrace{\mathbb{E}\left\|\sum_{j=0}^{k-1}(\partial F(X_j;\xi_j)-\partial f(X_j))\left(\frac{\mathbf{1}_n}{n}-W^{k-j-1}e_i\right)\right\|^2}_{=:T_2}$$

$$+2\gamma^2\underbrace{\mathbb{E}\left\|\sum_{j=0}^{k-1}\partial f(X_j)\left(\frac{\mathbf{1}_n}{n}-W^{k-j-1}e_i\right)\right\|^2}_{=:T_3}. \tag{11}$$

For $T_2$, we provide the following upper bounds:

$$T_2=\mathbb{E}\left\|\sum_{j=0}^{k-1}(\partial F(X_j;\xi_j)-\partial f(X_j))\left(\frac{\mathbf{1}_n}{n}-W^{k-j-1}e_i\right)\right\|^2$$

$$=\sum_{j=0}^{k-1}\mathbb{E}\left\|(\partial F(X_j;\xi_j)-\partial f(X_j))\left(\frac{\mathbf{1}_n}{n}-W^{k-j-1}e_i\right)\right\|^2$$

$$\leqslant\sum_{j=0}^{k-1}\mathbb{E}\|\partial F(X_j;\xi_j)-\partial f(X_j)\|^2\left\|\frac{\mathbf{1}_n}{n}-W^{k-j-1}e_i\right\|^2$$

$$\leqslant\sum_{j=0}^{k-1}\mathbb{E}\|\partial F(X_j;\xi_j)-\partial f(X_j)\|_F^2\left\|\frac{\mathbf{1}_n}{n}-W^{k-j-1}e_i\right\|^2$$

$$\overset{\text{(Lemma 4,Assumption 1-3)}}{\leqslant}n\sigma^2\sum_{j=0}^{k-1}\rho^{k-j-1}$$

$$\leqslant\frac{n\sigma^2}{1-\rho}.$$

For $T_3$, we provide the following upper bounds:

$$T_3=\mathbb{E}\left\|\sum_{j=0}^{k-1}\partial f(X_j)\left(\frac{\mathbf{1}_n}{n}-W^{k-j-1}e_i\right)\right\|^2$$

$$=\underbrace{\sum_{j=0}^{k-1}\mathbb{E}\left\|\partial f(X_j)\left(\frac{\mathbf{1}_n}{n}-W^{k-j-1}e_i\right)\right\|^2}_{=:T_4}$$

$$+\underbrace{\sum_{j\neq j'}\mathbb{E}\left\langle\partial f(X_j)\left(\frac{\mathbf{1}_n}{n}-W^{k-j-1}e_i\right),\partial f(X_{j'})\left(\frac{\mathbf{1}_n}{n}-W^{k-j'-1}e_i\right)\right\rangle}_{=:T_5}$$

To bound $T_3$ we bound $T_4$ and $T_5$ in the following: for $T_4$,

$$T_4 = \sum_{j=0}^{k-1} \mathbb{E} \left\| \partial f(X_j) \left( \frac{\mathbf{1}_n}{n} - W^{k-j-1} e_i \right) \right\|^2$$

$$\leqslant \sum_{j=0}^{k-1} \mathbb{E} \| \partial f(X_j) \|^2 \left\| \frac{\mathbf{1}_n}{n} - W^{k-j} e_i \right\|^2$$

$$\overset{\text{(Lemmas 4 and 5)}}{\leqslant} 3 \sum_{j=0}^{k-1} \sum_{h=1}^{n} \mathbb{E} L^2 Q_{j,h} \left\| \frac{\mathbf{1}_n}{n} - W^{k-j-1} e_i \right\|^2 + 3n\varsigma^2 \frac{1}{1-\rho}$$

$$+ 3 \sum_{j=0}^{k-1} \mathbb{E} \left\| \nabla f \left( \frac{X_j \mathbf{1}_n}{n} \right) \mathbf{1}_n^\top \right\|^2 \left\| \frac{\mathbf{1}_n}{n} - W^{k-j-1} e_i \right\|^2.$$

We bound $T_5$ using two new terms $T_6$ and $T_7$:

$$T_5 = \sum_{j \neq j'}^{k-1} \mathbb{E} \left\langle \partial f(X_j) \left( \frac{\mathbf{1}_n}{n} - W^{k-j-1} e_i \right), \partial f(X_{j'}) \left( \frac{\mathbf{1}_n}{n} - W^{k-j'-1} e_i \right) \right\rangle$$

$$\leqslant \sum_{j \neq j'}^{k-1} \mathbb{E} \left\| \partial f(X_j) \left( \frac{\mathbf{1}_n}{n} - W^{k-j-1} e_i \right) \right\| \left\| \partial f(X_{j'}) \left( \frac{\mathbf{1}_n}{n} - W^{k-j'-1} e_i \right) \right\|$$

$$\leqslant \sum_{j \neq j'}^{k-1} \mathbb{E} \| \partial f(X_j) \| \left\| \frac{\mathbf{1}_n}{n} - W^{k-j-1} e_i \right\| \| \partial f(X_{j'}) \| \left\| \frac{\mathbf{1}_n}{n} - W^{k-j'-1} e_i \right\|$$

$$\leqslant \sum_{j \neq j'}^{k-1} \mathbb{E} \| \partial f(X_j) \| \left\| \frac{\mathbf{1}_n}{n} - W^{k-j-1} e_i \right\| \| \partial f(X_{j'}) \| \left\| \frac{\mathbf{1}_n}{n} - W^{k-j'-1} e_i \right\|$$

$$\leqslant \sum_{j \neq j'}^{k-1} \mathbb{E} \frac{\| \partial f(X_j) \|^2}{2} \left\| \frac{\mathbf{1}_n}{n} - W^{k-j-1} e_i \right\| \left\| \frac{\mathbf{1}_n}{n} - W^{k-j'-1} e_i \right\|$$

$$+ \sum_{j \neq j'}^{k-1} \mathbb{E} \frac{\| \partial f(X_{j'}) \|^2}{2} \left\| \frac{\mathbf{1}_n}{n} - W^{k-j-1} e_i \right\| \left\| \frac{\mathbf{1}_n}{n} - W^{k-j'-1} e_i \right\|$$

$$\overset{\text{Lemma 4}}{\leqslant} \sum_{j \neq j'}^{k-1} \mathbb{E} \left( \frac{\| \partial f(X_j) \|^2}{2} + \frac{\| \partial f(X_{j'}) \|^2}{2} \right) \rho^{k - \frac{j+j'}{2} - 1}$$

$$= \sum_{j \neq j'}^{k-1} \mathbb{E} ( \| \partial f(X_j) \|^2 ) \rho^{k - \frac{j+j'}{2} - 1}$$

$$\overset{\text{Lemma 5}}{\leqslant} \underbrace{3 \sum_{j \neq j'}^{k-1} \left( \sum_{h=1}^{n} \mathbb{E} L^2 Q_{j,h} + \mathbb{E} \left\| \nabla f \left( \frac{X_j \mathbf{1}_n}{n} \right) \mathbf{1}_n^\top \right\|^2 \right) \rho^{k - \frac{j+j'}{2} - 1}}_{=: T_6}$$

$$+ \underbrace{\sum_{j \neq j'}^{k-1} 3n\varsigma^2 \rho^{k-1-\frac{j+j'}{2}}}_{=: T_7},$$

where $T_7$ can be bounded using $\varsigma$ and $\rho$:

$$T_7 = 6n\varsigma^2 \sum_{j > j'}^{k-1} \rho^{k-1-\frac{j+j'}{2}}$$

$$= 6n\varsigma^2 \frac{\left( \rho^{k/2} - 1 \right) \left( \rho^{k/2} - \sqrt{\rho} \right)}{\left( \sqrt{\rho} - 1 \right)^2 \left( \sqrt{\rho} + 1 \right)}$$

$$\leq 6n\varsigma^2 \frac{1}{\left(1 - \sqrt{\rho}\right)^2},$$

and we bound $T_6$:

$$T_6 = 3 \sum_{j \neq j'}^{k-1} \left( \sum_{h=1}^{n} \mathbb{E} L^2 Q_{j,h} + \mathbb{E} \left\| \nabla f \left( \frac{X_j \mathbf{1}_n}{n} \right) \mathbf{1}_n^\top \right\|^2 \right) \rho^{k - \frac{j+j'}{2} - 1}$$

$$= 6 \sum_{j=0}^{k-1} \left( \sum_{h=1}^{n} \mathbb{E} L^2 Q_{j,h} + \mathbb{E} \left\| \nabla f \left( \frac{X_j \mathbf{1}_n}{n} \right) \mathbf{1}_n^\top \right\|^2 \right) \sum_{j'=j+1}^{k-1} \sqrt{\rho}^{2k-j-j'-2}$$

$$\leq 6 \sum_{j=0}^{k-1} \left( \sum_{h=1}^{n} \mathbb{E} L^2 Q_{j,h} + \mathbb{E} \left\| \nabla f \left( \frac{X_j \mathbf{1}_n}{n} \right) \mathbf{1}_n^\top \right\|^2 \right) \frac{\sqrt{\rho}^{k-j-1}}{1 - \sqrt{\rho}}.$$

Plugging $T_6$ and $T_7$ into $T_5$ and then plugging $T_5$ and $T_4$ into $T_3$ yield the upper bound for $T_3$:

$$T_3 \leq 3 \sum_{j=0}^{k-1} \sum_{h=1}^{n} \mathbb{E} L^2 Q_{j,h} \left\| \frac{\mathbf{1}_n}{n} - W^{k-j-1} e_i \right\|^2$$

$$+ 3 \sum_{j=0}^{k-1} \mathbb{E} \left\| \nabla f \left( \frac{X_j \mathbf{1}_n}{n} \right) \mathbf{1}_n^\top \right\|^2 \left\| \frac{\mathbf{1}_n}{n} - W^{k-j-1} e_i \right\|^2$$

$$+ 6 \sum_{j=0}^{k-1} \left( \sum_{h=1}^{n} \mathbb{E} L^2 Q_{j,h} + \mathbb{E} \left\| \nabla f \left( \frac{X_j \mathbf{1}_n}{n} \right) \mathbf{1}_n^\top \right\|^2 \right) \frac{\sqrt{\rho}^{k-j-1}}{1 - \sqrt{\rho}}$$

$$+ \frac{3n\varsigma^2}{1 - \rho} + \frac{6n\varsigma^2}{\left(1 - \sqrt{\rho}\right)^2}$$

$$\leq 3 \sum_{j=0}^{k-1} \sum_{h=1}^{n} \mathbb{E} L^2 Q_{j,h} \left\| \frac{\mathbf{1}_n}{n} - W^{k-j-1} e_i \right\|^2$$

$$+ 3 \sum_{j=0}^{k-1} \mathbb{E} \left\| \nabla f \left( \frac{X_j \mathbf{1}_n}{n} \right) \mathbf{1}_n^\top \right\|^2 \left\| \frac{\mathbf{1}_n}{n} - W^{k-j-1} e_i \right\|^2$$

$$+ 6 \sum_{j=0}^{k-1} \left( \sum_{h=1}^{n} \mathbb{E} L^2 Q_{j,h} + \mathbb{E} \left\| \nabla f \left( \frac{X_j \mathbf{1}_n}{n} \right) \mathbf{1}_n^\top \right\|^2 \right) \frac{\sqrt{\rho}^{k-j-1}}{1 - \sqrt{\rho}}$$

$$+ \frac{9n\varsigma^2}{\left(1 - \sqrt{\rho}\right)^2},$$

where the last step we use the fact that $\frac{1}{1-\rho} \leq \frac{1}{(1-\sqrt{\rho})^2}$.

Putting the bound for $T_2$ and $T_3$ back to (11) we get the bound for $Q_{k,i}$:

$$Q_{k,i} \leq \frac{2\gamma^2 n \sigma^2}{1 - \rho} + 6\gamma^2 \sum_{j=0}^{k-1} \sum_{h=1}^{n} \mathbb{E} L^2 \left\| \frac{\sum_{i'=1}^{n} x_{j,i'}}{n} - x_{j,h} \right\|^2 \left\| \frac{\mathbf{1}_n}{n} - W^{k-j-1} e_i \right\|^2$$

$$+ 6\gamma^2 \sum_{j=0}^{k-1} \mathbb{E} \left\| \nabla f \left( \frac{X_j \mathbf{1}_n}{n} \right) \mathbf{1}_n^\top \right\|^2 \left\| \frac{\mathbf{1}_n}{n} - W^{k-j-1} e_i \right\|^2$$

$$+ 12\gamma^2 \sum_{j=0}^{k-1} \left( \sum_{h=1}^{n} \mathbb{E} L^2 \left\| \frac{\sum_{i'=1}^{n} x_{j,i'}}{n} - x_{j,h} \right\|^2 + \mathbb{E} \left\| \nabla f \left( \frac{X_j \mathbf{1}_n}{n} \right) \mathbf{1}_n^\top \right\|^2 \right) \frac{\sqrt{\rho}^{k-j-1}}{1 - \sqrt{\rho}}$$

$$+ \frac{18\gamma^2 n \varsigma^2}{(1 - \sqrt{\rho})^2}$$

$$\overset{\text{Lemma 4}}{\leqslant} \frac{2\gamma^2 n\sigma^2}{1-\rho} + \frac{18\gamma^2 n\varsigma^2}{(1-\sqrt{\rho})^2}$$

$$+ 6\gamma^2 \sum_{j=0}^{k-1}\sum_{h=1}^{n} \mathbb{E}L^2 Q_{j,h}\rho^{k-j-1}$$

$$+ 6\gamma^2 \sum_{j=0}^{k-1} \mathbb{E}\left\|\nabla f\left(\frac{X_j \mathbf{1}_n}{n}\right)\mathbf{1}_n^\top\right\|^2 \rho^{k-j-1}$$

$$+ 12\gamma^2 \sum_{j=0}^{k-1}\left(\sum_{h=1}^{n}\mathbb{E}L^2 Q_{j,h} + \mathbb{E}\left\|\nabla f\left(\frac{X_j \mathbf{1}_n}{n}\right)\mathbf{1}_n^\top\right\|^2\right) \frac{\sqrt{\rho}^{k-j-1}}{1-\sqrt{\rho}}$$

$$= \frac{2\gamma^2 n\sigma^2}{1-\rho} + \frac{18\gamma^2 n\varsigma^2}{(1-\sqrt{\rho})^2}$$

$$+ 6\gamma^2 \sum_{j=0}^{k-1} \mathbb{E}\left\|\nabla f\left(\frac{X_j \mathbf{1}_n}{n}\right)\mathbf{1}_n^\top\right\|^2 \left(\rho^{k-j-1} + \frac{2\sqrt{\rho}^{k-j-1}}{1-\sqrt{\rho}}\right)$$

$$+ 6\gamma^2 \sum_{j=0}^{k-1}\sum_{h=1}^{n} \mathbb{E}L^2 Q_{j,h}\left(\frac{2\sqrt{\rho}^{k-j-1}}{1-\sqrt{\rho}} + \rho^{k-j-1}\right). \tag{12}$$

Till now, we have the bound for $Q_{k,i}$. We continue by bounding its average $M_k$ on all nodes, which is defined by:

$$\mathbb{E}M_k := \frac{\mathbb{E}\sum_{i=1}^{n} Q_{k,i}}{n} \tag{13}$$

$$\overset{(12)}{\leqslant} \frac{2\gamma^2 n\sigma^2}{1-\rho} + \frac{18\gamma^2 n\varsigma^2}{(1-\sqrt{\rho})^2}$$

$$+ 6\gamma^2 \sum_{j=0}^{k-1} \mathbb{E}\left\|\nabla f\left(\frac{X_j \mathbf{1}_n}{n}\right)\mathbf{1}_n^\top\right\|^2 \left(\rho^{k-j-1} + \frac{2\sqrt{\rho}^{k-j-1}}{1-\sqrt{\rho}}\right)$$

$$+ 6\gamma^2 nL^2 \sum_{j=0}^{k-1} \mathbb{E}M_j\left(\frac{2\sqrt{\rho}^{k-j-1}}{1-\sqrt{\rho}} + \rho^{k-j-1}\right).$$

Summing from $k=0$ to $K-1$ we get:

$$\sum_{k=0}^{K-1} \mathbb{E}M_k \leqslant \frac{2\gamma^2 n\sigma^2}{1-\rho}K + \frac{18\gamma^2 n\varsigma^2}{(1-\sqrt{\rho})^2}K$$

$$+ 6\gamma^2 \sum_{k=0}^{K-1}\sum_{j=0}^{k-1} \mathbb{E}\left\|\nabla f\left(\frac{X_j \mathbf{1}_n}{n}\right)\mathbf{1}_n^\top\right\|^2 \left(\rho^{k-j-1} + \frac{2\sqrt{\rho}^{k-j-1}}{1-\sqrt{\rho}}\right)$$

$$+ 6\gamma^2 nL^2 \sum_{k=0}^{K-1}\sum_{j=0}^{k-1} \mathbb{E}M_j\left(\frac{2\sqrt{\rho}^{k-j-1}}{1-\sqrt{\rho}} + \rho^{k-j-1}\right)$$

$$\leqslant \frac{2\gamma^2 n\sigma^2}{1-\rho}K + \frac{18\gamma^2 n\varsigma^2}{(1-\sqrt{\rho})^2}K$$

$$+ 6\gamma^2 \sum_{k=0}^{K-1} \mathbb{E}\left\|\nabla f\left(\frac{X_k \mathbf{1}_n}{n}\right)\mathbf{1}_n^\top\right\|^2 \left(\sum_{i=0}^{\infty}\rho^i + \frac{2\sum_{i=0}^{\infty}\sqrt{\rho}^i}{1-\sqrt{\rho}}\right)$$

$$+ 6\gamma^2 nL^2 \sum_{k=0}^{K-1} \mathbb{E}M_k\left(\frac{2\sum_{i=0}^{\infty}\sqrt{\rho}^i}{1-\sqrt{\rho}} + \sum_{i=0}^{\infty}\rho^i\right)$$

$$\leqslant \frac{2\gamma^2 n\sigma^2}{1-\rho}K + \frac{18\gamma^2 n\varsigma^2}{(1-\sqrt{\rho})^2}K$$

$$+ \frac{18}{(1-\sqrt{\rho})^2}\gamma^2 \sum_{k=0}^{K-1} \mathbb{E}\left\|\nabla f\left(\frac{X_k \mathbf{1}_n}{n}\right)\mathbf{1}_n^\top\right\|^2$$

$$+ \frac{18}{(1-\sqrt{\rho})^2}\gamma^2 nL^2 \sum_{k=0}^{K-1} \mathbb{E}M_k,$$

where the second step comes from rearranging the summations and the last step comes from the summation of geometric sequences.

Simply by rearranging the terms we get the bound for the summation of $\mathbb{E}M_k$'s from $k=0$ to $K-1$:

$$\left(1 - \frac{18}{(1-\sqrt{\rho})^2}\gamma^2 nL^2\right)\sum_{k=0}^{K-1}\mathbb{E}M_k$$

$$\leqslant \frac{2\gamma^2 n\sigma^2}{1-\rho}K + \frac{18\gamma^2 n\varsigma^2}{(1-\sqrt{\rho})^2}K + \frac{18}{(1-\sqrt{\rho})}\gamma^2 \sum_{k=0}^{K-1}\mathbb{E}\left\|\nabla f\left(\frac{X_k\mathbf{1}_n}{n}\right)\mathbf{1}_n^\top\right\|^2$$

$$\implies \sum_{k=0}^{K-1}\mathbb{E}M_k \leqslant \frac{2\gamma^2 n\sigma^2}{(1-\rho)\left(1 - \frac{18}{(1-\sqrt{\rho})^2}\gamma^2 nL^2\right)}K + \frac{18\gamma^2 n\varsigma^2}{(1-\sqrt{\rho})^2\left(1 - \frac{18}{(1-\sqrt{\rho})^2}\gamma^2 nL^2\right)}K$$

$$+ \frac{18\gamma^2}{(1-\sqrt{\rho})^2\left(1 - \frac{18}{(1-\sqrt{\rho})^2}\gamma^2 nL^2\right)}\sum_{k=0}^{K-1}\mathbb{E}\left\|\nabla f\left(\frac{X_k\mathbf{1}_n}{n}\right)\mathbf{1}_n^\top\right\|^2. \qquad (14)$$

Recall (10) that $T_1$ can be bounded using $M_k$:

$$\mathbb{E}T_1 \leqslant \frac{L^2}{n}\sum_{i=1}^{n}\mathbb{E}Q_{k,i} = L^2\mathbb{E}M_k. \qquad (15)$$

We are finally able to bound the error by combining all above. Starting from (9):

$$\mathbb{E}f\left(\frac{X_{k+1}\mathbf{1}_n}{n}\right) \leqslant \mathbb{E}f\left(\frac{X_k\mathbf{1}_n}{n}\right) - \frac{\gamma - \gamma^2 L}{2}\mathbb{E}\left\|\frac{\partial f(X_k)\mathbf{1}_n}{n}\right\|^2 - \frac{\gamma}{2}\mathbb{E}\left\|\nabla f\left(\frac{X_k\mathbf{1}_n}{n}\right)\right\|^2$$

$$+ \frac{\gamma^2 L}{2n}\sigma^2 + \frac{\gamma}{2}\mathbb{E}T_1$$

$$\overset{(15)}{\leqslant} \mathbb{E}f\left(\frac{X_k\mathbf{1}_n}{n}\right) - \frac{\gamma - \gamma^2 L}{2}\mathbb{E}\left\|\frac{\partial f(X_k)\mathbf{1}_n}{n}\right\|^2 - \frac{\gamma}{2}\mathbb{E}\left\|\nabla f\left(\frac{X_k\mathbf{1}_n}{n}\right)\right\|^2$$

$$+ \frac{\gamma^2 L}{2n}\sigma^2 + \frac{\gamma}{2}L^2\mathbb{E}M_k.$$

Summing from $k=0$ to $k=K-1$ we get:

$$\frac{\gamma - \gamma^2 L}{2}\sum_{k=0}^{K-1}\mathbb{E}\left\|\frac{\partial f(X_k)\mathbf{1}_n}{n}\right\|^2 + \frac{\gamma}{2}\sum_{k=0}^{K-1}\mathbb{E}\left\|\nabla f\left(\frac{X_k\mathbf{1}_n}{n}\right)\right\|^2$$

$$\leqslant f(0) - f^* + \frac{\gamma^2 KL}{2n}\sigma^2 + \frac{\gamma}{2}L^2\sum_{k=0}^{K-1}\mathbb{E}M_k$$

$$\overset{(14)}{\leqslant} f(0) - f^* + \frac{\gamma^2 KL}{2n}\sigma^2$$

$$+\frac{\gamma}{2}L^2\frac{2\gamma^2n\sigma^2}{(1-\rho)\left(1-\frac{18}{(1-\sqrt{\rho})^2}\gamma^2nL^2\right)}K+\frac{\gamma}{2}L^2\frac{18\gamma^2n\varsigma^2}{(1-\sqrt{\rho})^2\left(1-\frac{18}{(1-\sqrt{\rho})^2}\gamma^2nL^2\right)}K$$

$$+\frac{\gamma}{2}L^2\frac{18\gamma^2}{(1-\sqrt{\rho})^2\left(1-\frac{18}{(1-\sqrt{\rho})^2}\gamma^2nL^2\right)}\sum_{k=0}^{K-1}\mathbb{E}\left\|\nabla f\left(\frac{X_k\mathbf{1}_n}{n}\right)\mathbf{1}_n^\top\right\|^2$$

$$=f(0)-f^*+\frac{\gamma^2KL}{2n}\sigma^2$$

$$+\frac{\gamma^3L^2n\sigma^2}{(1-\rho)\left(1-\frac{18}{(1-\sqrt{\rho})^2}\gamma^2nL^2\right)}K+\frac{9\gamma^3L^2n\varsigma^2}{(1-\sqrt{\rho})^2\left(1-\frac{18}{(1-\sqrt{\rho})^2}\gamma^2nL^2\right)}K$$

$$+\frac{9n\gamma^3L^2}{(1-\sqrt{\rho})^2\left(1-\frac{18}{(1-\sqrt{\rho})^2}\gamma^2nL^2\right)}\sum_{k=0}^{K-1}\mathbb{E}\left\|\nabla f\left(\frac{X_k\mathbf{1}_n}{n}\right)\right\|^2$$

By rearranging the inequality above, we obtain:

$$\implies \frac{\frac{\gamma-\gamma^2L}{2}\sum_{k=0}^{K-1}\mathbb{E}\left\|\frac{\partial f(X_k)\mathbf{1}_n}{n}\right\|^2+\left(\frac{\gamma}{2}-\frac{9n\gamma^3L^2}{(1-\sqrt{\rho})^2\left(1-\frac{18}{(1-\sqrt{\rho})^2}\gamma^2nL^2\right)}\right)\sum_{k=0}^{K-1}\mathbb{E}\left\|\nabla f\left(\frac{X_k\mathbf{1}_n}{n}\right)\right\|^2}{\gamma K}$$

$$\leqslant\frac{f(0)-f^*}{\gamma K}+\frac{\gamma L}{2n}\sigma^2+\frac{\gamma^2L^2n\sigma^2}{(1-\rho)\left(1-\frac{18}{(1-\sqrt{\rho})^2}\gamma^2nL^2\right)}+\frac{9\gamma^2L^2n\varsigma^2}{(1-\sqrt{\rho})^2\left(1-\frac{18}{(1-\sqrt{\rho})^2}\gamma^2nL^2\right)}.$$

which completes the proof. $\qquad\square$

***Proof to Corollary 2.*** Substitute $\gamma=\frac{1}{2L+\sigma\sqrt{K/n}}$ into Theorem 1 and remove the $\left\|\frac{\partial f(X_k)\mathbf{1}_n}{n}\right\|^2$ terms on the LHS. We get

$$\frac{D_1\sum_{k=0}^{K-1}\mathbb{E}\left\|\nabla f\left(\frac{X_k\mathbf{1}_n}{n}\right)\right\|^2}{K}$$

$$\leqslant\frac{2(f(0)-f^*)L}{K}+\frac{(f(0)-f^*)\sigma}{\sqrt{Kn}}+\frac{L\sigma^2}{4nL+2\sigma\sqrt{Kn}}$$

$$+\frac{L^2n}{(2L+\sigma\sqrt{K/n})^2D_2}\left(\frac{\sigma^2}{1-\rho}+\frac{9\varsigma^2}{(1-\sqrt{\rho})^2}\right)$$

$$\leqslant\frac{2(f(0)-f^*)L}{K}+\frac{(f(0)-f^*+L/2)\sigma}{\sqrt{Kn}}$$

$$+\frac{L^2n}{(\sigma\sqrt{K/n})^2D_2}\left(\frac{\sigma^2}{1-\rho}+\frac{9\varsigma^2}{(1-\sqrt{\rho})^2}\right).\qquad(16)$$

We first show $D_1$ and $D_2$ are approximately constants when (6) is satisfied.

$$D_1:=\left(\frac{1}{2}-\frac{9\gamma^2L^2n}{(1-\sqrt{\rho})^2D_2}\right),\quad D_2:=\left(1-\frac{18\gamma^2}{(1-\sqrt{\rho})^2}nL^2\right).$$

Note that

$$\gamma^2\leqslant\frac{(1-\sqrt{\rho})^2}{36nL^2}\implies D_2\geqslant1/2,$$

$$\gamma^2\leqslant\frac{(1-\sqrt{\rho})^2}{72L^2n}\implies D_1\geqslant1/4.$$

Since

$$\gamma^2\leqslant\frac{n}{\sigma^2K},$$

as long as we have

$$\frac{n}{\sigma^2 K} \leqslant \frac{(1 - \sqrt{\rho})^2}{36nL^2}$$

$$\frac{n}{\sigma^2 K} \leqslant \frac{(1 - \sqrt{\rho})^2}{72L^2 n},$$

$D_2 \geqslant 1/2$ and $D_1 \geqslant 1/4$ will be satisfied. Solving above inequalities we get (6).

Now with (6) we can safely replace $D_1$ and $D_2$ in (17) with $1/4$ and $1/2$ respectively. Thus

$$\frac{\sum_{k=0}^{K-1} \mathbb{E} \left\| \nabla f \left( \frac{X_k \mathbf{1}_n}{n} \right) \right\|^2}{4K}$$

$$\leqslant \frac{2(f(0) - f^*)L}{K} + \frac{(f(0) - f^* + L/2)\sigma}{\sqrt{Kn}}$$

$$+ \frac{2L^2 n}{(\sigma\sqrt{K/n})^2} \left( \frac{\sigma^2}{1 - \rho} + \frac{9\varsigma^2}{(1 - \sqrt{\rho})^2} \right). \tag{17}$$

Given (5), the last term is bounded by the second term, completing the proof. □

***Proof to Theorem 3.*** This can be seen from a simple analysis that the $\rho$, $\sqrt{\rho}$ for this $W$ are asymptotically $1 - \frac{16\pi^2}{3n^2}$, $1 - \frac{8\pi^2}{3n^2}$ respectively when $n$ is large. Then by requiring (6) we need $n \leq O(K^{1/6})$. To satisfy (5) we need $n \leq O\left(K^{1/9}\right)$ when $\varsigma = 0$ and $n \leq O(K^{1/13})$ when $\varsigma > 0$. This completes the proof. □

We have the following theorem showing the distance of the local optimization variables converges with a $O(1/K)$ rate, where the "$O$" swallows $n, \rho, \sigma, \varsigma, L$ and $f(0) - f^*$, which means it is safe to use any worker's local result to get a good estimate to the solution:

**Theorem 6.** *With $\gamma = \frac{1}{2L + \sigma\sqrt{K/n}}$ under the same assumptions as in Corollary 2 we have*

$$(Kn)^{-1}\mathbb{E} \sum_{k=0}^{K-1} \sum_{i=1}^{n} \left\| \frac{\sum_{i'=1}^{n} x_{k,i'}}{n} - x_{k,i} \right\|^2 \leqslant n\gamma^2 \frac{A}{D_2},$$

*where*

$$A := \frac{2\sigma^2}{1 - \rho} + \frac{18\varsigma^2}{(1 - \sqrt{\rho})^2} + \frac{L^2}{D_1} \left( \frac{\sigma^2}{1 - \rho} + \frac{9\varsigma^2}{(1 - \sqrt{\rho})^2} \right)$$

$$+ \frac{18}{(1 - \sqrt{\rho})^2} \left( \frac{f(0) - f^*}{\gamma K} + \frac{\gamma L\sigma^2}{2nD_1} \right).$$

Choosing $\gamma$ in the way in Corollary 2, we can see that the consensus will be achieved in the rate $O(1/K)$.

***Proof to Theorem 6.*** From (14) with $\gamma = \frac{1}{2L + \sigma\sqrt{K/n}}$ we have

$$\frac{\sum_{k=0}^{K-1} \mathbb{E}M_k}{K} \leqslant \frac{2\gamma^2 n\sigma^2}{(1 - \rho)D_2} + \frac{18\gamma^2 n\varsigma^2}{(1 - \sqrt{\rho})^2 D_2}$$

$$+ \frac{18\gamma^2}{(1 - \sqrt{\rho})^2 D_2} \frac{\sum_{k=0}^{K-1} \mathbb{E} \left\| \nabla f \left( \frac{X_k \mathbf{1}_n}{n} \right) \mathbf{1}_n^\top \right\|^2}{K}$$

$$= \frac{2\gamma^2 n\sigma^2}{(1 - \rho)D_2} + \frac{18\gamma^2 n\varsigma^2}{(1 - \sqrt{\rho})^2 D_2}$$

$$+ \frac{18\gamma^2 n}{(1 - \sqrt{\rho})^2 D_2} \frac{\sum_{k=0}^{K-1} \mathbb{E} \left\| \nabla f \left( \frac{X_k \mathbf{1}_n}{n} \right) \right\|^2}{K}$$

$$\overset{\text{Corollary 2}}{\leqslant} \frac{2\gamma^2 n\sigma^2}{(1-\rho)D_2} + \frac{18\gamma^2 n\varsigma^2}{(1-\sqrt{\rho})^2 D_2} + \frac{\gamma^2 L^2 n}{D_1 D_2}\left(\frac{\sigma^2}{1-\rho} + \frac{9\varsigma^2}{(1-\sqrt{\rho})^2}\right)$$

$$+ \frac{18\gamma^2 n}{(1-\sqrt{\rho})^2 D_2}\left(\frac{f(0)-f^*}{\gamma K} + \frac{\gamma L\sigma^2}{2nD_1}\right)$$

$$= \frac{n\gamma^2}{D_2} A.$$

This completes the proof. □

## Footnotes

[5]https://github.com/sixin-zh/mpiT.git