[Reviews · NeurIPS 2017]

Reviewer 1



This paper studies a decentralized parallel stochastic gradient descent (D-PSGD) and shows that decentralized algorithms might outperform centralized algorithms in this case. In this decentralized setting, every node performs SGD and exchanges information with neighbors using decentralized gradient descent (DGD). D-PSGD applies stochastic on the gradient information in DGD. With some assumptions on the functions, the authors show the theoretical result on when it converges. The numerical experiments comparing with C-PSGD is very interesting and promising. The stepsize depends on the number of iterations, which is mostly used in analysis of stochastic algorithms. A too small stepsize usually means a slow algorithm and a large stepsize will provide a solution that is too far from the optimum. Therefore, diminishing stepsizes are used in DGD and SGD, one question is what happens when the diminishing stepsize is used in D-PSGD.

Reviewer 2



The authors study the convergence rate of decentralized algorithms for parallel gradient descent methods. They show with their analysis that the decentralized methods have less communication complexity than the centralized methods. To support their analysis the authors also provide computational results, where they demonstrate that decentralized parallel algorithms run faster than the centralized parallel algorithms. The analysis is correct. However, it begs for more explanation. For instance, if I understand correctly, Corollary 2 states that K should be larger than the maximum of the two right-hand sides in relations (5) and (6). This implies that K may become too large and hence the step length may become very small. Then do we still have convergence with a constant step length? I may be missing a point here. Along this line, can the analysis be generalized to decreasing step lengths case? Can the results be improved if the authors concentrate on (strongly) convex objective functions?

Reviewer 3



In this paper, the authors present an algorithm for decentralized parallel stochastic gradient descent (PSGD). In contrast to centralized PSGD where worker nodes compute local gradients and the weights of a model are updated on a central node, decentralized PSGD seeks to perform training without a central node, in regimes where each node in a network can communicate with only a handful of adjacent nodes. While this network architecture has typically been viewed as a limitation, the authors present a theoretical analysis of their algorithm that suggests D-PSGD can achieve a linear speedup comparable to C-PSGD, but with significantly lower communication overhead. As a result, in certain low bandwidth or high latency network scenarios, D-PSGD can outperform C-PSGD. The authors validate this claim empirically. Overall, I believe the technical contributions of this paper could be very valuable. The authors claim to be the first paper providing a theoretical analysis demonstrating that D-PSGD can perform competitively or even outperform C-PSGD. I am not sufficiently familiar with the relevant literature to affirm this claim, but if true then I believe that the analysis provided by the authors is both novel and intriguing, and could have nontrivial practical impact on those training neural networks in a distributed fashion, particularly over high latency networks. The authors additionally provide a convincing experimental evaluation of D-PSGD, demonstrating the competitiveness of D-PSGD using modern CNN architectures across several network architectures that vary in scale, bandwidth and latency. I found the paper fairly easy to read and digest, even as someone not intimately familiar with the parallel SGD literature and theory. The paper does contain a number of small typographical errors that should be corrected with another editing pass; a small list of examples that caught my eye is compiled below. -- Minor comments -- Line 20: "pay" -> "paying" Line 23: "its" -> "their", "counterpart" -> "counterparts" Line 33: "parameter server" -> (e.g) "the parameter server topology" Algorithm 1: the for loop runs from 0 to K-1, but the output averages over x_{K}. Line 191: "on" -> "in" Line 220: "of NLP" -> "of the NLP experiment"